

# Warming effects of reduced sulfur emissions from shipping

Masaru Yoshioka[1], Daniel P. Grosvenor[1,2], Ben B. B. Booth[2], Colin P. Morice[2], and Ken S. Carslaw[1]

[1]Institute for Climate and Atmospheric Science, School of Earth and Environment, University of Leeds, Leeds, LS2 9JT, United Kingdom

[2]Met Office Hadley Centre, Exeter, EX1 3PB, United Kingdom

*Correspondence to*: Masaru Yoshioka (M.Yoshioka@leeds.ac.uk)

**Abstract.** The regulation introduced in 2020 that limits the sulfur content in shipping fuel has reduced sulfur emissions over global open oceans by about 80%. This is expected to have reduced aerosols that both reflect solar radiation directly and affect

cloud properties, with the latter also changing the solar radiation balance. Here we investigate the impacts of this regulation on aerosols and climate in the HadGEM3-GC3.1 climate model. The global aerosol effective radiative forcing caused by reduced shipping emissions is estimated to be 0.13 W m-2, which is equivalent to about 50% of the positive forcing caused by the global reduction in all anthropogenic aerosols since late 20th century. Ensembles of global coupled simulations from 2020-2049 predict a global mean warming of 0.04 K averaged over this period. Our simulations are not clear on whether the global

impact is yet to emerge or has already emerged because the present-day impact is masked by variability. Nevertheless, the impact of shipping emission reductions will have either already committed us to warming above the 1.5 K Paris target or will represent an important contribution that may help explain part of the rapid jump in global temperatures over the last 12 months. Consistent with previous aerosol perturbation simulations, the warming is greatest in the Arctic, reaching a mean of 0.15 K Arctic-wide and 0.3 K in the Atlantic sector of the Arctic (which represents greater than 10% increase in the total anthropogenic

warming since pre-industrial times).

## 1 Introduction

Globally ships emit around 10-13 Tg per year of sulfur dioxide ($SO_2$) into the atmosphere in recent years, which accounts for about 14% of global SO2 emissions from all sectors in both ECLIPSE (Klimont et al., 2017) and CEDS (Hoesly et al., 2018) datasets. In the atmosphere, SO2 is oxidised to form sulfate, which either condenses on the existing aerosol particles or forms

new particles, and hence contributes additional aerosol mass and number. These particles directly modify the Earth's energy budget by scattering solar and terrestrial radiation and indirectly affect it through changing the cloud microphysical (droplet numbers and sizes affecting the reflectivity) and macrophysical (cloud cover, height, liquid and ice water paths) properties. Over the ocean, due to the dark surface (low albedo) any change in aerosol and cloud reflectivity can potentially have a large impact on the Earth's energy budget.




Ship exhausts are known to form ship tracks, which are the linear features of enhanced cloudiness or cloud brightness up to hundreds of kilometres in length that are sometimes clearly visible in satellite images, typically in the regions of marine stratocumulus clouds (e.g., Conover, 1966; Coakley et al., 1987; Toll et al., 2019; Diamond et al., 2020). Although the majority of ocean-going ships do not leave identifiable ship tracks, the sulfur species will still be widely dispersed and potentially cause

significant, but less apparent, aerosol-cloud interactions that modulate the Earth's energy budget (Possner et al., 2018). Particulate matter originating from shipping emissions causes substantial air pollution in coastal areas of the world, causing an estimated 400,000 premature deaths every year (Sofiev et al., 2018). To mitigate this, the United Nation's International Maritime Organisation (IMO) set Sulfur Emission Control Areas (SECAs) in inland seas in Northern Europe and along the coasts of North America in which sulfur emissions from shipping were limited by specifying a maximum fuel sulfur content

of 0.1% by mass. In addition, from January 2020, the IMO imposed further restricted the maximum fuel sulfur content of ships in all ocean regions outside the SECAs to 0.5% of fuel mass. It is claimed that this will prevent about 600,000 premature deaths in the coming years (Corbett et al., 2016). This is a large step change from the previous regulation that allowed fuel sulphur contents of up to 3.5% that will substantially affect shipping sulfur emissions and potentially atmospheric composition and climate.


To investigate the effects of the new emission regulation on atmospheric composition and the responses of the climate, we performed two ensembles of coupled climate model simulations with and without the sudden emission reduction due to the IMO regulation change after 2020.

## 50 2 Methods

### 2.1 Model

We use the HadGEM3-GC3.1-LL (also called HadGEM3 N96ORCA1; Kuhlbrodt et al., 2018), the low-resolution version of HadGEM3 Global Coupled version 3.1 model (Williams et al., 2017), where the atmosphere model with 1.875° x 1.25° horizontal resolution and 85 vertical levels is coupled with the 1° resolution NEMO ocean model (Madec et al., 2017). The

atmosphere model involves the UKCA chemistry-aerosol scheme (O'Connor et al., 2014) which includes the GLOMAP-mode two moment aerosol model (Mann et al., 2010). Black carbon (BC), organic carbon (OC), sea salt (SS) and sulfate (SU) aerosols are simulated in GLOMAP-mode where microphysical interactions between different aerosol species and sizes are represented. Mineral dust is included separately in the CLASSIC bin scheme (Bellouin et al., 2007; Woodward, 2001). $SO_2$ is oxidised to form sulphuric acid via the gas-phase reaction with OH radicals in the troposphere or through the aqueous-

phase reactions with $O_3$ and $H_2O_2$ in cloud droplets. Gas phase sulfuric acid then either condenses on the existing aerosol particles or forms new particles through binary homogeneous nucleation throughout the atmosphere (Vehkamäki et al., 2002) or through organically mediated nucleation in the boundary layer (Metzger et al., 2010). In this model, oxidant concentrations





are prescribed and do not change by these reactions. 2.5% of the SO2 from both anthropogenic and natural sources is assumed to be emitted as primary aerosol particles to represent subgrid scale oxidation and condensation.

## 2.2 Experimental design

We set up two 35-year ensembles of simulations between 2015 and 2049 that differ in the change of shipping SO2 emissions after 2020. Concentrations of well-mixed greenhouse gases and reactive gases including oxidants were prescribed following the ScenarioMIP SSP1-2.6 scenario (O'Neill et al., 2016).

Our base case scenario generally follows the ECLIPSE v6b scenario (Klimont et al., 2017), but with a small modification. In ECLIPSE v6 the global shipping SO2 emissions fall from 10.1 Tg SO2 (14% of anthropogenic emissions) in 2015 to 2.1 Tg (4%) in 2020, consistent with the sulfur emission reduction by the IMO 2020 regulation. However, in our simulations, we repeated the 2015 emissions for 2016-2019, instead of smoothly interpolating the ECLIPSE v6b values between 2015 and 2020, to represent the sudden reduction due to the regulation change in year 2020. We call this scenario SHIP20 because it includes the reduction of shipping SO2 emissions to 20% of its pre-2020 value. In the other (counterfactual) scenario, we repeated the shipping SO2 emissions of 2015 until the end of the simulation. We call this SHIP100. Figure 1 shows the emission pathways in both of these scenarios as well as the difference between them in space and time.

We used the ECLIPSE v6b scenario for emissions of primary carbonaceous aerosols (black carbon and organic carbon) from anthropogenic and biofuel sources. Emissions of primary carbonaceous aerosols from biomass burning, volcanic emissions of SO2, as well as biogenic monoterpenes (a precursor gas of secondary organic aerosol) were also prescribed according to the SSP1-2.6 scenario. Emissions of marine dimethyl sulfide (DMS), a precursor of sulfate aerosol, are calculated interactively within the model (Mulcahy et al., 2020) as a function of surface wind speeds and prescribed surface seawater DMS concentrations given by Lana et al. (2011). Sea salt emissions are calculated interactively within the model using wind speeds over the sea.

The emission reductions of various aerosols and precursor gases due to COVID-19 pandemic were not included and are not expected to be significant on the decadal time scales of interest here.

Twelve pairs of simulations were created under the two emission scenarios each starting from slightly different initial conditions taken from the HadGEM3-LL Coupled Model Intercomparison Project (CMIP6; Eyring et al., 2016) historic simulations. By creating paired simulations in two ensembles we aim to preclude sampling bias caused by the choice of initial conditions. The use of ensembles of 35-year simulations allows us to examine the transient response of the climate system.



**Figure 1: SO2 emissions used in the simulations. (a) Global annual SO2 emissions by sector in the SHIP20 scenario from 2015 to 2050 based on the ECLIPSE v6b dataset. (b) Global monthly shipping SO2 emission pathways in the SHIP100 (blue) and SHIP20 (red) scenarios for the same period. (c) Change in SO2 emissions in 2020 (SHIP20 minus SHIP100 emissions). Global aerosol emissions from transport, waste, and flaring are included in the dataset but can hardly be seen in (a) due to their relatively minor contributions, although they can be regionally important.**



### 2.3 Estimation of temperature changes from pre-industrial baseline

Global and regional mean temperature changes between pre-industrial (PI) and present-day (PD) were estimated using the PI temperatures of the 16-member CMIP6 UKESM1.0 ensembles (Sellar et al., 2019; Senior et al., 2020) as a baseline, which were calculated as the ensemble mean of the CMIP6 historic simulations for the three-year period of 1850-1852. However,

there are differences in global and regional mean PD temperatures between the CMIP6 and our simulations, just before the new shipping regulation came into effect. Therefore, temperature changes from the PI to a given year in the post-2020 period were estimated using the CMIP6 PI baseline and our simulated temperatures adjusted for the differences in 2017-2019 mean temperatures between CMIP6 and our ensembles (before any difference in simulations is made due to shipping emission change between two scenarios) in both the SHIP100 and SHIP20 scenarios.

### 2.4 Estimation of effective radiative forcing

Aerosol effective radiative forcing (ERF) was estimated from global atmosphere-only UKESM1 simulations nudged to ERA-Interim analyses data (Berrisford et al., 2011). Two simulations were conducted for the period of nine years from 2015 to 2023, with shipping emissions held constant at 2015 and 2020 levels, corresponding to SHIP100 and SHIP20 scenarios, respectively. The simulation data from the last eight years were utilized, with the data from 2015 excluded. The ERF was

calculated following the methodology outlined in Ghan (2013) and by comparing the results obtained from the two simulations.

### 3. Results

Figure 2 shows the difference in ensemble-mean sulfate aerosol column burden (vertically integrated mass per unit area) between the SHIP100 and SHIP20 scenarios averaged over the entire simulation period. The global mean reduction is 0.14 mg m-2, corresponding to 4.6% of the case in SHIP100. The spatial pattern of changes in sulfate burden largely follows the

pattern of emissions, but with greater spreading due to transport of SO2 and the resulting aerosol. The strongest reductions can be seen in the large coastal region of Southeast Asia followed by in the Mediterranean and around the Arabian Peninsula. Relatively large reductions are seen over the large region covering the Eastern tropical Atlantic, Europe and North Africa, tropical Indian Ocean and West Pacific.



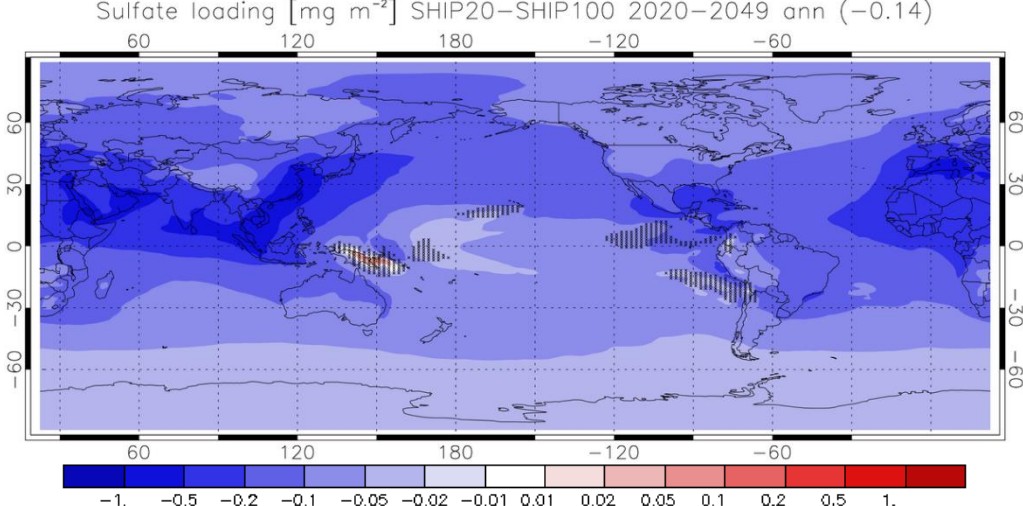

**Figure 2. Change in ensemble mean sulfate aerosol column burden [mg m–2] from SHIP100 to SHIP20. The differences are statistically significant at a 95% confidence level in paired t-test everywhere *except* the hatched regions.**

Figure 3 shows the aerosol ERF (SHIP20-SHIP100) caused by the reduction in sulfur emissions quantified from global atmosphere-only nudged simulations for 2016-2023. The global annual mean aerosol ERF is 0.128 W m-2, with an interannual standard deviation of 0.016. Strong positive ERF can be seen extending from N Indian Ocean through SE Asia to China, along the N Pacific shipping corridor from Japan, and around Iberian Peninsula and Morocco, consistent with the reduced sulfur emission in Figure 1 and sulfate burden in Figure 2.

Although the emission reductions equate to only about 14% of global SO2 emissions from all sectors, this forcing is about 50% of that caused by reductions in all anthropogenic aerosol emissions since the 1990s when the magnitude of the negative global aerosol forcing peaked. This is based on the weighted historical timeseries of CMIP6 forcings in Smith et al. (2021), which estimates a 1990-2020 aerosol ERF of 0.25 W m-2.



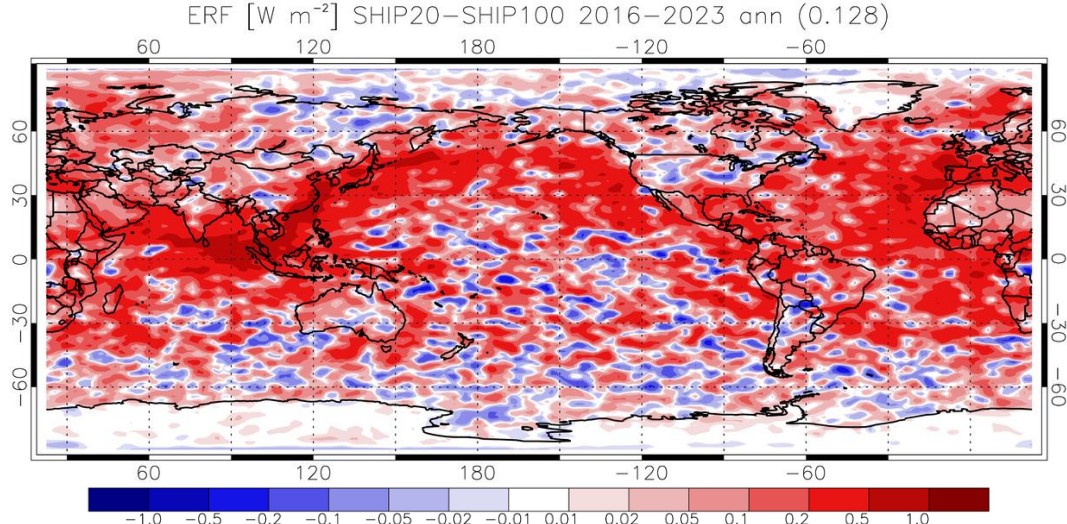


**Figure 3. Effective radiative forcing (W m–2) from shipping sulfur reduction (SHIP20 – SHIP100), calculated from atmosphere-only nudged simulations. The plot has been smoothed by averaging each grid box value with the values from its neighbouring grid boxes.**


Figure 4 shows the global map of the difference in annual mean 1.5 m temperatures between the two scenarios (SHIP20-SHIP100) averaged over three 10-year periods: 2020-2029, 2030-2039 and 2040-2049. Figures S1 and S2 show the same, but for December-February and June-August. These plots show statistically significant warming in the 2030s and 2040s in the tropical eastern Indian Ocean to western Pacific Ocean region, around the Mediterranean, in eastern North America, and in the

Atlantic Ocean north of 60°N. Although these locations are not necessarily consistent in these two decades, many of these regions correspond to the regions with relatively strong positive ERFs.

An interesting feature of the distribution of temperature changes is a warming in the tropical eastern Indian Ocean to western Pacific Ocean and a cooling in the central to eastern Pacific. This warming and cooling pattern corresponds to the pattern of increased and decreased rainfall in the tropics between 90°E and 135°W (Figure 5). These resemble the anomalous patterns

seen during La Niña. Figures S3 and S4 show that this pattern corresponds to changes in top-of-atmosphere longwave flux and high-cloud amount in the simulations. Furthermore, Figure S5 indicates that these changes are associated with the strengthening of Walker Circulation, with enhancements of convergence of low to mid-level horizontal wind around 125°E (top panel), upward motion over the western Pacific (100-125°E) and downward motion over the central Pacific (130°E-160°W; bottom panel) in the SHIP20 ensemble. The increase in high-cloud amount is hypothesized to be due to the increased

upward moisture transport over the western Pacific (not shown) caused by the enhanced upward motion and increased convection. A unique feature of reduced aerosol column burden is seen over New Guinea in Figure 2, which is likely due to increased rainfall there. Together with the ocean processes such as changes in surface currents and the zonal SST gradient in the equatorial Pacific (not shown), the mechanism causing the La Niña-like condition is consistent with the positive Bjerknes feedback (Bjerknes, 1969; Rädel et al., 2016) that explains how the ENSO anomaly is reinforced.







**Figure 4. Differences in time-averaged ensemble annual mean 1.5 m temperatures between the SHIP20 and SHIP100 simulations in the 2020s (top), 2030s (middle), and 2040s (bottom). The hatching shows where there the differences between the two ensembles are statistically significant at a 95% confidence level in paired t-test. DJF and JJA seasonal averages are shown in Figures S1 and S2.**



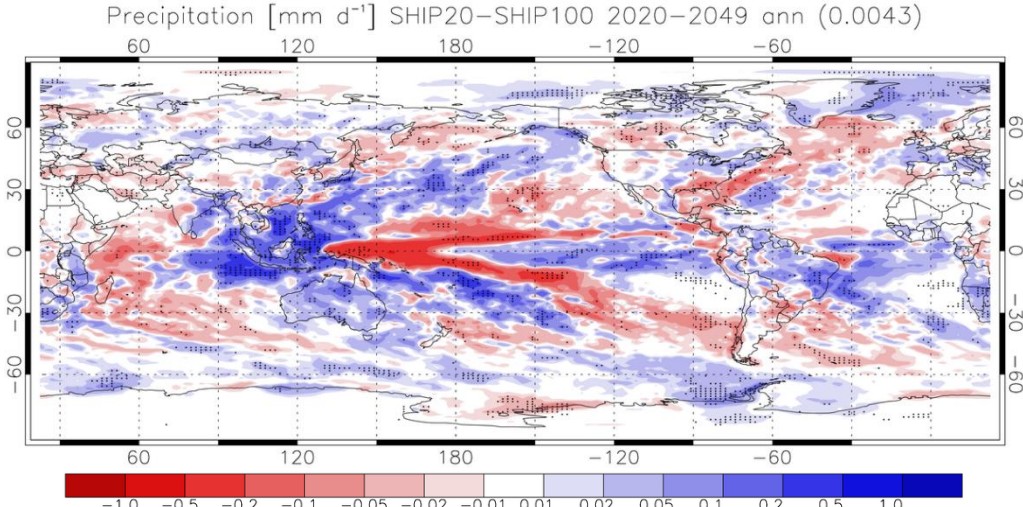

**Figure 5. Differences in ensemble mean annual precipitation between SHIP20 and SHIP100 simulations averaged for 2020-2049.**

Figure 6 shows the global time evolution of the ensemble-mean warming in the two scenarios compared to the pre-industrial baseline as well as measures of statistical significance (standard errors and p-values in paired t-tests). According to this, the global annual mean warming exceeds 1.5 K around 2024 regardless of the shipping emissions change. However, the reduction

in shipping sulfur emissions (SHIP20 compared to SHIP100) causes additional warming starting in the late 2020s. In the following decades the additional warming by the shipping emissions reduction is 0.04-0.05K and is statistically significant (p=0.001 in 2030s and 0.007 in 2040s). It also suggests that the shipping emissions reduction could be an important factor that determines whether global warming reaches 2.0 K in the 2040s. Seasonally, the warming is larger and more significant in northern summer than in winter.




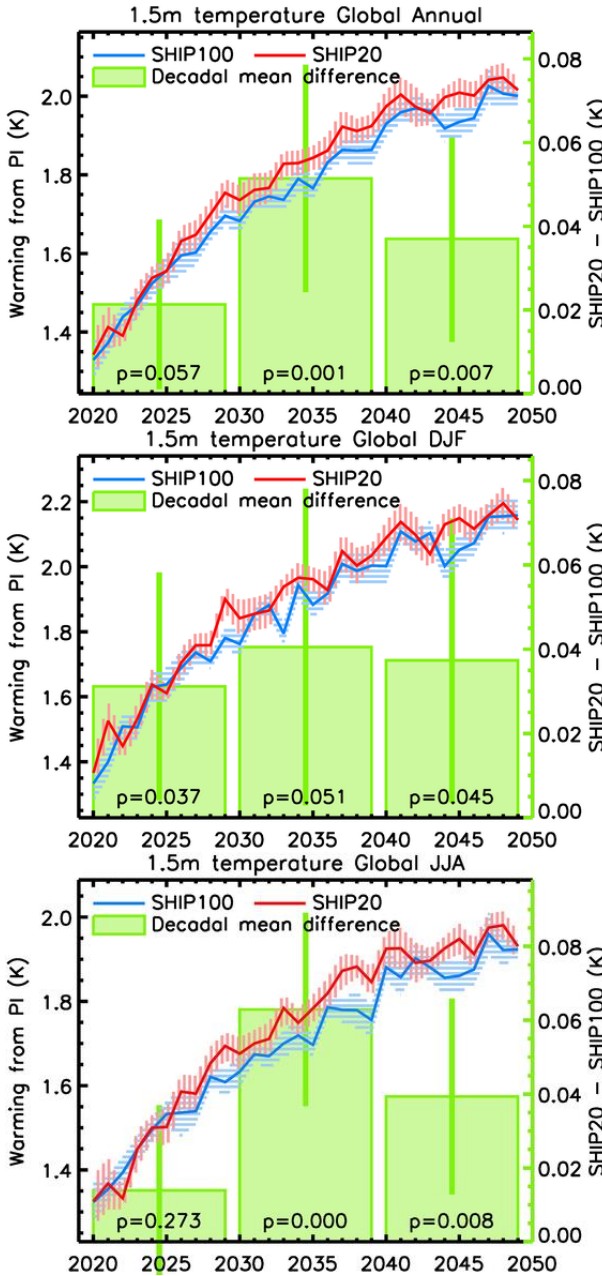

**Figure 6. The evolution of global averaged ensemble mean 1.5 m air temperatures in the SHIP20 (red line) and SHIP100 (blue line) simulations compared to the 1850-1852 means. The shading around the lines shows the one standard error range. The green rectangles show the differences in the decadal means (secondary y-axis) and their one standard error range (green vertical lines) for annual (top), DJF (middle) and JJA (bottom) data. P-values in paired t-tests for the decadal means are shown near the bottom of the corresponding green rectangles.**





Figure 7 shows the regional time evolutions of the ensemble-mean warmings for the Atlantic, tropical Indian and Pacific
      Oceans, northern North America, Europe, the Arctic and the Atlantic sector of the Arctic.  Figures S6 and S7 show the same
      things but for DJF and JJA.





**Figure 7. Same as Figure 6 but for regional annual averages. From top left to bottom right, Atlantic (50S-0; 45W-15E and 0-50N; 70-10W), Tropical Indian and Pacific Oceans (15S-20N; 45-150E), Northern North America (40-70N; 135-60W), Europe and Mediterranean (30-70N; 10W-30E), Arctic (60-90N, 180W-180E), and Atlantic sector of Arctic (60-80N; 90W-30E). DJF and JJA averages are shown in Figure S6 and S7.**



## 4. Discussion and conclusions

Our results suggest that how we experience global warming in the next few decades will be dependent on both the climate impact of SO2 shipping cuts and natural variability (Figures 6 and 7) as well as the underlying greenhouse gas driven climate change. At the regional scale (Figure 7) the magnitude of the climate impact appears to emerge and sometimes reduce again, through time. This suggests that we would need a larger ensemble size to more fully isolate the climate change signal (or potentially interesting dynamical feedbacks which have yet to be identified). The simulations show that the climate impact of

SO2 cuts takes a few years to emerge, which is consistent with the climate response to other step changes in forcings in other contexts. Andrews et al. (2019) shows that the UKESM1 model, used in this study, realises 44%, 59% and 68% the longer-term climate response within 5, 10 and 20 years of a step change in forcing, respectively. However, given the effect of sub-decadal variability on the signal in other parts of the record, caution is needed as this may alternatively reflect variability masking the initial response. As such, it is unclear whether we would expect the real world to already be experiencing the

warming impact of SO2 shipping cuts, or whether the signal will emerge in the next few years, which has implications for interpreting their impact.

If the global climate impact of SO2 shipping cuts will emerge in the next few years, as our Figure 6 results suggest, then this has consequences for our ability to achieve our global warming targets. Whilst the global temperature impact is modest in the context of longer-term global warming, ranging from 0.04 K to 0.05 K (Figure 6), it becomes more relevant when we consider

global targets of 1.5 or 2 K. 2023 is estimated to have been around 1.45 K warmer than average conditions of 1850-1900 (World Meteorological Organization, 2024). Annual temperatures are subject to year-to-year variability, with El Niño conditions contributing to 2023 temperatures, hence long-term warming estimates often use longer averaging periods or other methods to filter out the effects of such variability. The IPCC AR6 reported observed warming based on the most recent 10-year average. An update to IPCC AR6 diagnostics reported the 2013-2022 decade at 1.14 [0.9 to 1.4] degrees above 1850-

1900 (Forster et al. 2023). On this basis, 0.04 K warming from shipping would represent 11% of the remaining warming 1.5 degrees from the 2013-2022 decade. More instantaneous measures estimate warming to 2022 at around 1.26 K, based on attributable warming estimates (Forster et al., 2023) and a combination of observations with model projections (Betts et al., 2023). For these estimates of warming to 2022, an additional 0.04 K warming from shipping SO2 reductions would account for almost 17% of the remaining warming to 1.5 K.

Both scenarios (Figure 6) suggest that (in the absence of dramatic immediate CO2 emission cuts) we will exceed 1.5 K in the next couple of years, so the SO2 commitment to exceeding 1.5 K may have contextual value, only. The SO2 cuts are likely to impact 2 K targets more meaningfully as they narrow the window before we realise this level of global warming to 17 (rather than 23 years) under this emission scenario. Consequently, in the absence of immediate cuts to all greenhouse gas emissions, the recent SO2 cuts may have already made 1.5 K and 2 K harder to achieve.

What is the role of these SO2 cuts in the exceptional recent warming record? 2023 was 0.17 K warmer than the previous record year (2016), even though 2016 represented an exceptionally strong El Niño, and we do not expect to experience the full impact





of the emerging 2023 El Niño until 2024. The 0.17 K warming between the last two years with El Niños (which occurred 7 years apart) would represent a considerable acceleration of global warming if this were caused by greenhouse gas driven climate change alone. Cuts of SO2 from shipping and the impact of water vapour injection into the stratosphere by the Hunga
Tonga-Hunga Ha'apai (HTHH) volcanic eruption represent two factors that may help to explain at least part of this warming. The HTHH eruption may have contributed up to 0.04 K global warming (Jenkins et al, 2023) because, unusually, it contributed a large stratospheric water vapour injection that was counterbalanced by a more modest sulphate aerosol injection (Zhu et al, 2022). Its net warming or cooling impact is still contested (with Schoeberl et al, 2023 arguing that it represented a net cooling) but the HTHH eruption could be another factor that influenced the warm 2023 temperatures. Together, the effect of shipping
emission reductions (if they have emerged) and the eruption could explain up to 0.08 K of the 0.17 K warming since 2016. The difference between the two is that we would expect any warming from the HTHH eruption to rapidly decay (the e-folding timescale of volcanic global temperature impact is roughly 2.5 years) whereas the additional warming from SO2 shipping cuts is expected to persist.

There are similar challenges in interpreting the spatial temperature impacts of marine SO2 emission cuts (Figure 4), with
inconstancy in the ensemble mean pattern evident from one decade to the next. Whilst we cannot rule out the potential role of interesting dynamical feedbacks, this may just reflect the need to deploy larger climate model ensembles to estimate the climate change signal and that these decade-to-decade changes reflect variability superimposed on this underlying pattern. There are, however, inferences of consistent changes that can be drawn from the spatial patterns in Figure 4. One of these inferences is that marine SO2 cuts produce a Pacific SST pattern that looks like a "Central Pacific La Niña" pattern (Capotondi et al, 2015)
or La Niña Modoki pattern (Cai and Cowan 2009), with cooler central Pacific temperatures with warmer temperatures to north, south, west and in this case east. Unlike La Niña patterns arising naturally from variability, this pattern is associated with a net global warming, but we would still expect that the Pacific SST gradients associated with this pattern would similarly project on to wider regional climate with similar effects and via similar mechanisms. Whilst ENSO variability continues to superimpose onto future Pacific SSTs, the impact of the marine SO2 cut preconditions the mean SST states, which we can
expect to similarly precondition regional ENSO driven impacts on decadal timescales.

In wider regions, the limits of our ensemble size appear to limit our ability to isolate the climate change pattern alone. However, shipping SO2 cuts do lead to consistent warming in many regions, even when there is variability in both the patterns (Figure 4) and timeseries (Figure 7). Shipping cuts cause marked Arctic amplification and warming in the Atlantic, Indian and Pacific Oceans. There are suggestions that this warming also influences continental conditions, such as NW North America, India
and East Africa. The possibility that it may have been a factor, albeit a small one, that preconditioned the temperature extremes experienced in N America in 2021 is an intriguing one but will require further work and experiments that are beyond the scope of this paper.

The 2020 rapid cuts in shipping SO2 emissions are likely to have had a long-term climate impact, influencing both global and regional warming as well as changing regional preconditions to how we experience climate variability over the next twenty to
thirty years.

## Code and data availability

ECLIPSE V6b dataset can be downloaded from https://iiasa.ac.at/models-tools-data/global-emission-fields-of-air-pollutants-and-ghgs. Simulation data and codes required to reproduce the main figures in this article will be provided on Zonodo (link

placeholder).

## Author contributions

MY, KC, and DG designed the study. MY prepared input data for the models, set up the models, ran the simulations, processed and analyzed the simulation data, and prepared the figures. KC, DG, and BB provided advice on how to proceed with the study. The manuscript was written by MY and BB, and all co-authors contributed to discussions and suggestions in finalizing

the manuscript.

## Acknowledgments

MY, KC, and DG would like to acknowledge funding from the NERC ACRUISE grant (NE/S004807/1). This work used Monsoon2, a collaborative High-Performance Computing facility funded by the Met Office and the Natural Environment Research Council. This work used JASMIN, the UK collaborative data analysis facility. MY, KC and DG would like to

express their gratitude to Laura Wilcox for providing insights on a part of the study. MY would like to acknowledge Klimont Zbigniew for providing ECLIPSE 6b dataset and to Chris Smith and Steven Turnock for providing support in processing ECLIPSE data. MY would also like to express his gratitude to Mohit Dalvi, Martin Andrews, Rosalyn Hatcher and Grenville Lister for providing support in model setups and resolving troubles in model runs.

## Competing interests

KC is a member of the editorial board of Atmospheric Chemistry and Physics.

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
