# Peer review of "Warming effects of reduced sulfur emissions from shipping"

_EGUsphere, 2024_

## Referee Comment (RC2)

Review of "Warming effects of reduced sulfur emissions from shipping"
Yoshita et al.

The authors report on the warming observed in a global climate model when regulated reductions in sulfur emissions from shipping are prescribed. These reductions are compared against "business as usual" sulfur emissions, showing a global mean average warming of 0.04 K and interesting regional differences in radiative forcing and temperature responses. Regional differences were partly attributed to dynamic changes to atmospheric circulation and precipitation patterns. Overall, the paper is well-written and provides an interesting investigation into the radiative response of changing global aerosol emissions in a global climate model and its implications for future warming. The paper is well suited for ACP and I have only a few comments to be addressed in a minor revision.

- Introduction, Lines 33-35: I think an appropriate reference to work on "invisible ship tracks" could be included here, see (Manshausen et al., 2022).

- General figure comment: the authors should add letters to plots that have multiple panels so they can be clearly identified and referenced in the main text. There are many such instances where data in a specific panel is referenced, but the reader is required to hunt for that panel because it is not specified (see some specifics in the following comments).

- Line 103: Are you missing the citations Sellar et al. (2019) and Senior et al. (2020)? I don't see these citations in the reference list.

- Continuing the point in the previous comment, I am particularly interested in how the baseline simulations and PI temperatures were estimated. Can the authors please speak more to these simulations, the uncertainty, and their comparison to other CMIP6 historic simulations (and correct the citations)?

- Line 115: The citation Ghan (2013) is not provided in the reference list. Given that the ERF is a key feature of this paper, I ask that the authors please provide this citation and explain the methodology used.

- Line 120: "The spatial pattern of changes in sulfate burden largely follows the pattern of emissions..." please reference Fig. 1c at the end of this statement.

- Line 134: "…consistent with the reduced sulfur emission in Figure 1…" please reference panel c of Fig. 1.

- Line 151: "…correspond to the regions with relatively strong positive ERFs." Please reference Fig. 3.

- Line 183: "Seasonally, the warming…" After looking at Figure 6 for several moments, I finally noticed that the left ordinate has different maximum limits for

each panel. Is there a specific reason this was done? Further to that point, the authors argue that the warming is "larger and more significant" in NH summer than winter. For this statement, the seasonal difference in warming should be quantified in magnitude and with statistical significance. Are the authors able to attribute the difference to specific processes? Does the seasonal difference relate at all to the change in circulation/precipitation patterns discussed earlier in this section?

- Line 209-211: This statement about "climate responses to other step changes in forcings" is rather vague. Can the authors please provide clarify with a few examples of such changes/responses with citation?

**References**

Manshausen, P., Watson-Parris, D., Christensen, M., Jalkanen, J., & Stier, P. (2022). Invisible ship tracks show large cloud sensitivity to aerosol. *Nature, 610*(7930), 101-+. Article.

---

## Author Comment (AC2)

SW TOA flux [W m⁻²] SHIP20−SHIP100 2020−2049 djf (0.13)

SW TOA flux [W m⁻²] SHIP20−SHIP100 2020−2049 jja (0.20)

Positive downward (warming)

---

## Author Response (AR1)

Response to reviewer 1:

*Using the HadGEM3-GC3.1 climate model, the authors investigate the impacts of reduced shipping SO₂ emissions. They estimate the aerosol effective radiative forcing caused by this reduction to be 0.13 W/m². Ensembles of global coupled simulations from 2020-2049 predict a global mean warming of 0.04 K averaged over this period. The authors suggest that the impact of shipping emission reductions could represent a significant contribution to the rapid global temperature rise observed from 2022 to 2023. While the authors have conducted extensive simulations and applied various methods to reach their conclusions, there are several concerns regarding the numerical design and data analysis.*

*My major concern is with the methodology used to adjust variations in temperature from the 1850s to the 2020s using differences in global annual temperature between CMIP6 PD and PI. Given the significant internal biases among models, this approach seems unreasonable. I recommend that the authors run additional PI simulations using the HadGEM3-GC3.1 model to estimate PD-PI differences and then compare these values with the CMIP6 datasets.*

Following the reviewer's comment, we have switched to CMIP6 historical ensemble with HADGEM3 GC3.1 (rather than UKESM1), the same model used in this study. Gillet et al. (2021) shows this ensemble produces PI to PD warming close to the middle of the multi-model and observed ranges. Since we used the simulation results for 2013-2015 from this ensemble as the initial conditions for our simulations, the adjustments previously applied to the pre-2020 climate are no longer necessary. Even though this is a small ensemble with only 4 members, by using the long term mean for 1850-1900 as the PI baseline, the uncertainty from internal variability should have been minimised. This change has only a small effect on global warming predictions, although relatively large changes can be seen regionally.

*Additionally, there are concerns about several other conclusions, some of which appear misleading:*

1. ***Aerosol Effective Radiative Forcing****: The statement that "the global aerosol effective radiative forcing caused by reduced shipping emissions is estimated to be 0.13 W/m², which is equivalent to about 50% of the positive forcing caused by the global reduction in all anthropogenic aerosols since the late 20th century" is misleading. Reductions in anthropogenic emissions since the late 20th century have contributed to warming primarily due to a decrease in scattering aerosols, and to cooling due to absorbing aerosols like soot. This statement could be misinterpreted to suggest that half of the aerosol warming effect is from shipping reductions alone.*

With this statement we did not intend to imply that half of the aerosol warming effect caused by aerosol reductions after the late 20[th] Century period was from the shipping reductions, but rather that we estimate that the shipping emissions likely caused an additional warming that is equivalent to 50% of the net warming from the historical aerosol reductions after the late 20[th] Century. We have refined the abstract to make this clearer as follows:
"The global aerosol effective radiative forcing caused by reduced shipping emissions is estimated to be 0.13 W m⁻², which is equivalent to an additional ~50% to the net positive forcing resulting from the reduction in all anthropogenic aerosols from the late 20[th] century to the pre-2020 era."

2. ***Temperature and Precipitation Changes (Figures 4 and 5)****: The authors discuss changes in temperature and precipitation, but only a few areas show statistical significance. As a result, the discussion about the warming trend lacks robustness.*

As the reviewer points out, Figure 4 shows statistically significant warming only in small regions. However, when we take global decadal mean temperatures, we obtain statistically significant differences between the two ensembles in the 2030s and 2040s as indicated by the p-values and the green error bars shown in figure 6. We also found statistically significant temperature differences in several regions and decades in figure 7. We discuss the warming effects mainly based on these figures rather than figures 4 and 5.

3. ***Walker Circulation (Figure Analysis)****: The authors note a reduction in LW at the TOA in the western tropical Pacific Ocean and attribute this to a strengthening of the Walker Circulation. However, they do not discuss changes in SW, pressure gradients, or sea surface temperature, which are necessary to support this claim.*

We thank the reviewer for highlighting this. We have updated the manuscript to provide a fuller picture. The patterns in SSTs are strongly correlated with the 1.5m temperature (shown in the manuscript) over the ocean (which is why we have not explicitly shown SSTs separately). The change in SW radiative flux is dominant in the net radiative balance at the TOA globally and in most regions (see Figure S4, which has been revised following this comment), which is expected given the aerosol driver, but the region around the maritime continent and tropical Pacific are unique in that LW change dominates, as shown in Figure S4. We wanted to keep the focus on LW in the main manuscript and included additional text as well as plots for SW and net flux changes (top and bottom panels) to Figure S4 to provide this fuller picture in the supplementary information.

4. ***Temperature Comparison (2023 vs. 2016)****: The comparison of global temperatures in 2023 to 2016 is problematic. Although 2016 was a strong El Niño year, it does not necessarily represent the highest global mean temperature in the past decade. Moreover, the conclusion that "the effect of shipping emission reductions (if they have emerged) and the HTHH eruption could explain up to 0.08 K of the 0.17 K warming since 2016" lacks sufficient evidence and requires further clarification.*

We appreciate this comment, as it has prompted us to clarify our thinking. Tackling the first part of this comment first (whether 2023 and 2016 is a reasonable comparison). We wanted to encapsulate just how large the global temperature increase was in 2023. We accept the reviewer's point, that different global temperature records either attribute 2016 or 2020 as the previous warmest year (due to the margins of error). Different El Nino in 2016 and 2023 also leaves some ambiguity as an indicator of underlying climate change trend (because both El Nino events are unlikely to have represented the same magnitude of enhanced global temperature). To qualify this sentence in the revised manuscript, we now reference the WMO assessment of 2023 temperatures (see copied and pasted text, below). This now states the WMO estimates for the two previous record years (2016 and 2020) based on all available global temperature records. This text now qualifies the 2016 to 2023 trend based on the WMO estimate from all reconstructions (recognising that it is not an unambiguous estimate of the underlying warming) and now also includes the Dunstone et al, 2023 estimate of the warming not explained by global warming and ENSO variability (an additional +0.1–0.12°C that is unexplained). This revised text now reads as follows:

*"What is the role of these SO2 cuts in the exceptional recent warming record? 2023 was recorded as 1.45 ± 0.12K above the pre-industrial era, which smashed the previous record years, 2016 and 2020 at 1.29 ±0.12K and 1.27 ±0.12K, respectively (WMO, 2024). The emergence of El Nino in 2023 is likely to have contributed but is unlikely to explain the magnitude of the 2023 increase. Whilst not unambiguous, the 2016 (strong El Niño) to 2023 (emerging El Nino) trend of 0.16°C would represent a considerable acceleration of global warming if this were caused by greenhouse gas driven climate change alone. Dunstone et al, 2024 estimate that there is likely to be an unexplained +0.1–0.12K to the 2023 temperatures, not explained by global warming and ENSO variability."*

In response to the 2[nd] part of the reviewer's comment, we have sought to qualify and clarify the conclusions drawn in this section. We have done this by first citing Dunstone et al, 2024 who identified both the HTHH eruption and SO2 shipping cuts as potential explanations, where we first introduce this in the paragraph. We have expanded the discussion to provide a more nuanced discussion – that now (a) identifies the 0.1 to 0.12K of unexplained warming in 2024 (Dunstone et al, 2024), (b) notes that if HTHH was on the upper end of the published estimates and if warming from shipping cuts has already emerged they might combine to explain 0.08k of this unexplained warming and (c) it is possible that neither of these factors may be playing a role in this unexplained warming. The text now reads:

*"could be a potential factor that may have influenced the warm 2023 temperatures. Our 0.04K estimate of additional warming from SO2 shipping cuts provides a quantitative estimate that goes beyond Dunstone et al, 2024. If the contribution of the HTHH is on the upper end of published estimates and if the warming effect of SO2 shipping cuts have emerged, then they could potentially combine to explain up to 0.08K of the 0.1k to 0.12K of unexplained 2023 warming identified in Dunstone et al, 2024. The difference between the two contributors is that we would expect any warming from the HTHH eruption to rapidly decay (the e-folding timescale of volcanic global temperature impact is roughly 2.5 years) whereas the additional warming from $SO_2$ shipping cuts is expected to persist. However, if the HTHH temperature contribution was more modest (or even negative) and/or warming from SO2 shipping cuts have not emerged then we need to look for other potential explanations (perhaps indicating a marked acceleration of global warming). Given the large unexplained warming in 2023, it is important that we do not dismiss SO2 cuts as a potential explanatory factor, given credible evidence from the experiments presented here, that such cuts are capable of affecting the global temperature record"*

**Reviewer 2:**

*Review of "Warming effects of reduced sulfur emissions from shipping" Yoshita et al. The authors report on the warming observed in a global climate model when regulated reductions in sulfur emissions from shipping are prescribed. These reductions are compared against "business as usual" sulfur emissions, showing a global mean average warming of 0.04 K and interesting regional differences in radiative forcing and temperature responses. Regional differences were partly attributed to dynamic changes to atmospheric circulation and precipitation patterns. Overall, the paper is well-written and provides an interesting investigation into the radiative response of changing global aerosol emissions in a global climate model and its implications for future warming. The paper is well suited for ACP and I have only a few comments to be addressed in a minor revision.*

- *Introduction, Lines 33-35: I think an appropriate reference to work on "invisible ship tracks" could be included here, see (Manshausen et al., 2022).*

Reference added.

- General figure comment: the authors should add letters to plots that have multiple panels so they can be clearly identified and referenced in the main text. There are many such instances where data in a specific panel is referenced, but the reader is required to hunt for that panel because it is not specified (see some specifics in the following comments).

Letters were added to each panel.

- Line 103: Are you missing the citations Sellar et al. (2019) and Senior et al. (2020)? I don't see these citations in the reference list.

Sorry, we accidentally submitted the first manuscript with incomplete reference list. However, these references are no longer necessary because we do not use UKESM1 data (see the response to the next comment).

- Continuing the point in the previous comment, I am particularly interested in how the baseline simulations and PI temperatures were estimated. Can the authors please speak more to these simulations, the uncertainty, and their comparison to other CMIP6 historic simulations (and correct the citations)?

Following the reviewer's comment, we have switched to CMIP6 historical ensemble with HADGEM3 GC3.1 (rather than UKESM1), the same model used in this study. Gillet et al. (2021) shows this ensemble produces PI to PD warming close to the middle of the multi-model and observed ranges. Since we used the simulation results for 2013-2015 from this ensemble as the initial conditions for our simulations, the adjustments previously applied to the pre-2020 climate are no longer necessary. Even though this is a small ensemble with only 4 members, by using the long term mean for 1850-1900 as the PI baseline, the uncertainty from internal variability should have been minimised. This change has a minimum effect on global warming predictions, although relatively large changes can be seen regionally.

- Line 115: The citation Ghan (2013) is not provided in the reference list. Given that the ERF is a key feature of this paper, I ask that the authors please provide this citation and explain the methodology used.

The reference is now included.
Ghan, S. J.: Technical Note: Estimating aerosol effects on cloud radiative forcing, Atmos. Chem. Phys., 13, 9971–9974, https://doi.org/10.5194/acp-13-9971-2013, 2013.

- Line 120: "The spatial pattern of changes in sulfate burden largely follows the pattern of emissions..." please reference Fig. 1c at the end of this statement.

'(Figure 1c)' was added.

- Line 134: "…consistent with the reduced sulfur emission in Figure 1…" please reference panel c of Fig. 1.

Reference added.

- Line 151: "…correspond to the regions with relatively strong positive ERFs." Please reference Fig. 3.

Reference added.

- Line 183: "Seasonally, the warming…" After looking at Figure 6 for several moments, I finally noticed that the left ordinate has different maximum limits for each panel. Is there a specific reason this was done? Further to that point, the authors argue that the warming is "larger and more significant" in NH summer than winter. For this statement, the seasonal difference in warming should be quantified in magnitude and with statistical significance. Are the authors able to attribute the difference to specific processes? Does the seasonal difference relate at all to the change in circulation/precipitation patterns discussed earlier in this section?

The reviewer is right. Since we do mention the seasonal differences, using the same y ranges makes more sense. The global figures are now updated. On the other hand, y-axes in the regional figures (Figure 7, S6 and S7) are still optimised for individual plots because using the same y-axes for annual, DJF and JJA makes it hard to see the warming effects in some cases. We do not get into the details of seasonal analyses in regions.
Seasonal means between two scenarios and p-values for 2030s and 2040s were added in the text. Globally, SW warming dominates LW cooling due to reduced aerosol ERF (which is negative in SW and positive in LW). Aerosol ERF is stronger in NH (Figure 3) due to the higher shipping emissions (Figure 2). The SW radiative effect is stronger in northern summer when solar radiation is more intense in NH (see the figure below). Although there should be complex interactions in the climate system, radiation alone can explain the stronger warming in northern summer.
The following text was added. "This is likely due to the large reduction of aerosol loading in Northern Hemisphere having stronger effects in northern summer, when the solar radiation is more intense."

[Figure]

SW TOA flux [W m⁻²] SHIP20–SHIP100 2020–2049 djf (0.13)

SW TOA flux [W m⁻²] SHIP20–SHIP100 2020–2049 jja (0.20)

- Line 209-211: This statement about "climate responses to other step changes in forcings" is rather vague. Can the authors please provide clarify with a few examples of such changes/responses with citation?

Thank you for this comment. The reference and discussion in the following sentence were intended to clarify this with examples, but we accept that this was not clear. To address this comment, we have expanded the following discussion as follows (which now more clearly links to wider CMIP responses, identifies a figure where this is explicitly shown and uses "for example" to link this discussion with this previous statement):

"*The simulations show that the climate impact of SO2 cuts takes a few years to emerge, which is consistent with the climate response to other step changes in forcings in other contexts. For example, Figure 1b in Andrews et al. (2019) shows the global temperature response to a step change in CO2 (instantaneous quadrupling in this case) for a wide range of CMIP models (including the UKESM1 model used in this study). This shows that UKESM realises 44%, 59% and 68% the longer-term climate response within 5, 10 and 20 years of a step change in forcing, respectively.*"

---

## Author Response (AR2)

**Reviewer #1**

The authors utilize the HadGEM3-GC3.1 climate model to examine the impacts of reduced $SO_2$ emissions from shipping. They estimate the resulting aerosol effective radiative forcing to be 0.13 W/m². Additionally, their global coupled simulations for the period 2020-2049 predict a global mean warming of 0.04 K, averaged across this timeframe. The revised manuscript shows significant improvement and addresses the comments I raised in my previous review. I recommend acceptance of the paper after minor revisions.

1. Why does sulfate reduction lead to warming in the tropical eastern Indian Ocean, yet cooling in the central to eastern Pacific? How might this scenario change if sulfate reduction were not implemented? What specific role does $SO_2$ from shipping play in this process? Moreover, do emissions from dimethyl sulfide (DMS) and sea salt also contribute to these effects?

Sulfate reduction causes reduction in negative aerosol ERF due to aerosol-radiation interaction and aerosol-cloud interactions, causing warming. Warming near the surface of eastern Indian Ocean and western Pacific enhances the upward motion in these regions and easterly anomaly in the trade wind in the central Pacific as shown in Figure S5. This resembles the La Nina and leads to westward anomaly of surface current and enhanced upwelling of sea water to cool SST in this region where the impact of reduced aerosol is small (as inferred by Figure 1 and 3). Figure S6 is added to show the westward anomaly of surface current in the tropical Pacific. This is our interpretation and it is briefly outlined in section 3, although this is not the central point of this study.

We are evaluating the change due to the reduction of shipping sulfur emissions by comparing two ensembles under SHIP20 and SHIP100 scenarios. If sulfate reduction is not implemented, there will be no difference in any variable because we are comparing identical scenarios.

SO2 from shipping is quickly oxidised to form sulfate particles, which exert negative radiative forcing mainly through the aerosol-cloud interactions. It is this radiative forcing that caused the changes described above.

The emissions of DMS and sea salt are calculated interactively in the model based on surface windspeed. However, we do not expect significant differences in their emissions caused by shipping sulfate reduction. Therefore, their effects are largely cancelled out in the differences between ensembles, resulting in minimal net effect and a small contribution to the uncertainty due to the random variability.

2. The study primarily discusses the changes in shortwave and longwave radiative forcing to explain the observed temperature changes. However, sensible heat and latent heat fluxes are also important components that could influence temperature variations. Including these metrics would provide a more complete explanation of the underlying mechanisms driving temperature change.

The focus of this paper is radiative effects of reduced aerosols due to shipping fuel change and their impacts on temperature. There is a well-established physical relationship between radiative energy imbalances and global mean temperature without the need to explore sensible and latent heat

fluxes. An analysis of sensible and latent heat would require a full exploration of all the relevant regional energy terms as in Richardson et al. (2018) in the PDRMIP study (https://journals.ametsoc.org/view/journals/clim/31/23/jcli-d-17-0240.1.xml), including the additional terms that they included. This is vastly beyond the scope of our study, which aims only to explore the temperature responses.

3. Line 162: There appears to be an issue with Figure S5, as it only contains one plot. Please check and correct this figure.

We appreciate that the reviewer pointed out that this part of texts refers to wrong figures. Instead of correcting the text we changed the order of figures in supplementary materials because it is the order in which they are referred to in the main text. The old Figure S3 is now Figure S5 that is referred to here. The text three lines above referring to Figures S3 and S4 were also previously wrong but now they refer to the correct figures.

4. Lines 165-167: In Figure 2, why does increased precipitation correspond with higher sulfate loading? This relationship requires further clarification.

We thank the reviewer to find an error in the text here as well. This text should state that reduced rainfall (shown in figure 5) caused the increase in aerosol loading (as shown in figure 2) due to reduced wet deposition. The text has been corrected.

**Reviewer #2**

accepted as is